# Novel Findings regarding the Bioactivity of the Natural Blue Pigment Genipin in Human Diseases

**DOI:** 10.3390/ijms23020902

**Published:** 2022-01-14

**Authors:** Magdalena Bryś, Karina Urbańska, Beata Olas

**Affiliations:** 1Department of Cytobiochemistry, Faculty of Biology and Environmental Protection, University of Lodz, Pomorska 141/3, 90-236 Lodz, Poland; magdalena.brys@biol.uni.lodz.pl; 2Faculty of Medicine, Medical University of Lodz, 90-419 Lodz, Poland; karina.urbanska@stud.umed.lodz.pl; 3Department of General Biochemistry, Faculty of Biology and Environmental Protection, University of Lodz, Pomorska 141/3, 90-236 Lodz, Poland

**Keywords:** genipin, blue colorant, biological activity, safety

## Abstract

Genipin is an important monoterpene iridoid compound isolated from *Gardenia jasminoides* J.Ellis fruits and from *Genipa americana* fruits, or genipap. It is a precursor of a blue pigment which may be attractive alternative to existing food dyes and it possesses various potential therapeutic properties such as anti-cancer, anti-diabetic and hepatoprotective activity. Biomedical studies also show that genipin may act as a neuroprotective drug. This review describes new aspects of the bioactivity of genipin against various diseases, as well as its toxicity and industrial applications, and presents its potential mechanism of action.

## 1. Introduction

*Gardenia jasminoides* J.Ellis is a popular shrub in the Rubiaceae family, which naturally grows in the laurel forests of China, Japan, Taiwan or Vietnam. Its fruits are well known, being called *Zhizi* in China and *Sanshishi* in Japan, where they are used as a natural colorant and play an important role in traditional medicine [1]. *G. jasminoides* fruit extract has been used as an effective oral treatment for hepatic disorders and inflammation in traditional Chinese medicine [1,2]. Furthermore, preparations containing *G. jasminoides* fruit extract have also been found to have good therapeutic effects on diseases of the central nervous system, such as cerebral stroke, dementia, and depression [2]. Phytochemicals from medicinal plants often offer a better and safer substitute to synthetic medication [3]. *G. jasminoides* fruits are also used as functional food supplements in China and East Asia, and the flowers as a food ingredient and dietary supplement, for example, in tea.

About 160 chemical compounds have been described from this plant. Of these, the primary bioactive compounds are its iridoid glycosides and yellow pigment, which demonstrate various biological activities. The most widely used iridoids isolated from the *G. jasminoides* fruit are genipin and geniposide [4,5]; however, the iridoid glycoside content can vary by 5–6% between different regions [6].

The hydrolysis of geniposide and gardenoside by the β-glucosidase results in the production of genipin, a water-soluble iridoid monoterpenoid whose maximum absorbance (496 nm) does not change with the pH of the environment. Genipin can be reacted with amino acids such as glycine, lysine or phenylalanine to obtain the blue dye gardenia blue (Figure 1) [4,5,6,7]. Studies indicate that genipin is absorbed via the intestine and transported to the liver through the portal bloodstream [8]. Although various methods exist for isolating genipin from *G. jasminoides* fruits, large-scale isolation is usually performed microbiologically with *Penicillum nigrans*; the fungus hydrolyzes the geniposide into aglycone genipin by β-glucosidase [9,10]. More details are described in review paper of Ramos-de-la-Pena [10].

Gardenia blue is not resistant to temperatures greater than 80 °C; however, it does resist changes in environmental pH. It can thus be used for coloring both liquid products and solid products, including jellies and candies; however, dyeing jellies or candies brings with it a change of color to blue–green [4]. A similar colorant, gardenia yellow, including carotenoids obtained from the *G. jasminoides* fruits also has a long history of use in Asian countries. It has low allergenicity and stable chemical properties, it has antioxidant properties and is non-toxic; however, it is not yet permitted as a food colorant in the European Union and the USA [1].

Recently, two new iridoid glycosides (genipin 1,10-di-*O*-α-l-rhamnoide, and genipin 1,10-di-*O*-β-d-xylopyranoside) were isolated from gardenia fructus, the dried fruits of *G. jasminoides* [11]. The authors note that these compounds may act as tyrosinase inhibitors and antioxidants.

Genipin may be also obtained from *Genipa americana* L, a plant growing in Central South America, southern Mexico, and the Caribbean. The fruit, known as genipap, chipara, maluco, jaguar, huito, and caruto, is 8–10 cm long, with an elliptical shape. The sources of genipin and the conversion process of gardenia colorants (gardenia yellow and gardenia blue) are presented in Figure 1.

As well as being a blue food colorant, in vitro and in vivo studies have found gardenia to have various biological properties, including antioxidant, anti-inflammatory and anticancer activity. These play important roles in its mechanisms of action and its beneficial effects on the cardiovascular, nervous and digestive systems, among others (Table 1) [1,2,8,12,13,14,15]. This review reviews the current state of knowledge regarding the bioactivity of genipin, i.e., the aglycone of geniposide, in various diseases. This review is based on studies identified in electronic databases, including PubMed, Web of Knowledge, Scopus and ScienceDirect. The last search was run on December 15, 2021. The following terms were used: “genipin” and “blue colorant”.

## 2. Genipin in Food Products

Genipin from *G. jasminoides* is available for use in food products in East Asia, including Korea and Japan. This compound, as a blue colorant, obtained from *G. americana* or in combination with other fruits, is commercially exploited in European Union and the USA as concentrated fruit juices; as a colorant, it is exempt from certification [6,14]. More details about factors affecting the formation and stability of genipin are described by Neri-Numa et al. [14] The use of genipin in food products is summarized in Table 1.

## 3. Hepatoprotective Properties

The hepatoprotective effect of genipin was first noted by Jing-hua Peng [17]. Since then, genipin was found to enhance the distribution of multidrug resistance-associated protein 2 (Mrp2) to the bile canaliculi and promote mRNA and protein synthesis in hepatocytes, potentiating bilirubin disposal in vivo [18]. Elsewhere, genipin (25–200 mg/kg) was found to significantly decrease mortality and serum aminotransferase activity and attenuate the apoptosis of hepatocytes induced by D-galactosamine/lipopolysaccharide (LPS) [19]; it was also observed to reduce lipid peroxidation in a mouse model.

Genipin (50 mg/kg) has also been found to decrease oxidative stress, including lipid peroxidation, and apoptosis in mice [20]. Similarly, based on a study of C57BL/6 mice, Wang et al. [21] found that an intravenous injection of genipin (2.5 mg/kg) may protect against hepatotoxicity induced by carbon tetrachloride (CCl_4_); this protection may be facilitated by attenuating the inflammatory response and inducing autophagy, which may be mediated by the mTOR (mechanistic target of rapamycin) and p38 MAPK (mitogen-activated protein kinase) signaling pathways. Seo et al. [22] also note that 25, 50, and 100 mg/kg genipin seems to prevent D-galactosamine and LPS-induced hepatic injury in mice through suppression of necroptosis-mediated inflammasome signaling.

In a mouse model, genipin effectively antagonized high-fat diet-induced hyperlipidemia and hepatic lipid accumulation by regulating the miR-142a-5p/SREBP-1c axis (sterol regulatory element-binding protein 1c) [23]. The authors observe that obese mice treated with genipin (5 and 20 mg/kg per day) demonstrated a decrease in body weight, serum lipid levels and hepatic lipid accumulation.

The hepatoprotective properties of genipin may also be based on the action of uncoupling protein 2 (UCP2), mitochondria quality regulation, anti-fibrinogenic activity, and amelioration of hepatic steatosis. However, (I) most studies on this area were conducted in vitro and on animal models, (II) the optimum effective dose of genipin in clinical application remains unknown, and (III) its selectivity and specificity for specific targets are relatively low. The therapeutic potential of genipin in various liver diseases is given in more detail by Jing-hua Peng [17] and Fan et al. [8]

Sohn et al. [24] report that genipin has protective effects on gastrointestinal disorders such as gastritis and gastric ulcers in male rats: when administered at 50 and 100 mg/kg, genipin demonstrated an inhibitory effect against indomethacin-induced gastric ulcers and HCl/ethnol-induced acute gastritis.

## 4. Neuroprotective Properties

A key element of progression in neurodegenerative diseases, such as Alzheimer’s disease and Parkinson’s disease, is the development of an inflammatory response in the brain [25]. For example, higher levels of proinflammatory cytokines, such as interleukin (IL)-1β and IL-6, and tumor necrosis factor-α (TNF-α), have been observed in the blood, brain, and cerebral spinal fluid in patients with Alzheimer’s disease and Parkinson’s. Such chronic inflammation may induce the production of free radicals, synapse dysfunction, and promote the formation of amyloid plaque and Lewy bodies. Therefore, anti-inflammatory factors, including chemical compounds, may play an important role in the prevention and treatment of neurodegenerative diseases [2]. Recent studies based on animal models and cell lines have found genipin to display anti-inflammatory potential, potentially by inhibiting nitric oxide synthase, nitric oxide production and NF-κB (nuclear factor kappa B) expression; however, it may act through other mechanisms. For example, genipin was found to inhibit the LPS-induced inflammatory response in BV2 microglial cells by activating the Nrf2 (nuclear factor erythroid 2-related factor 2) signaling pathway, in vitro and in vivo. Moreover, genipin (≤ 20 µM) did not demonstrate any cytotoxicity; it also inhibited certain inflammatory mediators, such as NO (nitric oxide), prostaglandin E_2_ (PGE_2_), TNF-α and Il-1β, in a concentration-dependent manner, i.e., at 5, 10, and 20 µM [26].

Recently, in an in vitro model based on human neuroblastoma SH-SY5Y cells, Lin et al. [27] found that the conjugation of tacrine with genipin induces autophagy against Alzheimer’s disease, and that it enhances effects on acetylcholinesterase in vitro. The conjugate also appears to be a good modulator of the expression of p53 and Bcl-2 (B-cell lymphoma 2). However, the authors do not report the concentration of the conjugate used in the study.

The incidence of neurodegenerative diseases is higher in people with insulin dysregulation or insulin resistance. For example, Zhang et al. [28] note that genipin inhibits UCP2-mediated proton leak and acutely reverses beta-cell dysfunction caused by obesity and high glucose levels in isolated pancreatic islets. This is a significant finding because UCP2 negatively regulates glucose-stimulated insulin secretion.

In addition, an important component of neurological illness is chronic cerebral ischemia. Zhao et al. [29] note that genipin (50 mg/kg daily for 3 days) was also found to protect against cerebral ischemia-reperfusion injury in adult male C57BL/6 mice in vivo. In this case, genipin was believed to act by regulating the UCP2-SIRT3 (sirtuin 3) signaling pathway.

Various papers indicate that genipin may have anti-depressive properties, but the possible mechanisms are not fully understood. It has been proposed that genipin can increase the concentration of a brain-derived neutrotropic factor in the hippocampus, exerting its effect through the monoaminergic neurotransmitter system [30,31].

More information on the pharmacological actions of genipin for the treatment of neurodegenerative diseases of the central nervous system, including Alzheimer’s disease and Parkinson’s disease, and its anti-depressive activity is given in a review by Li et al. [2].

## 5. The Molecular Basis of the Antitumor Activity of Genipin—An Update

The molecular basis of the anti-tumor activity of genipin has been extensively characterized in two review articles by Habtemariam and Lentini [32] and Shanmugam et al. [33]. The papers provide a detailed description of the role of genipin in the inhibition of various stages of the neoplastic process and discuss its potential use in the treatment of various cancers. The reviews summarize the processes as diagrams. However, research has moved on since the publication of these articles and brought with it significant new findings; the evidence base now encompasses a growing number of cells from various cancers [32].

Genipin was found to demonstrate antitumor activity at all stages of the carcinogenesis process. It appears to limit cell growth and cell-cycle progression in bladder and colon cancer cells. For example, genipin was found to cause cell-cycle arrest at the G0/G1-phase and deregulation of cell cycle regulators in bladder cancer T24 and 5637 cells, to inhibit the viability and growth of bladder cancer cells and to inhibit the growth of T24 xenograft tumors [34,35].

Under the influence of genipin, bladder cancer cell cultures demonstrated an increase in the number of apoptotic cells, loss of mitochondrial membrane potential, translocation of Bax (Bcl-2 associated X) protein into mitochondria, and release of cytochrome c. Moreover, the levels of PI3K (phosphoinositide-3 kinase) and Akt phosphorylation were significantly decreased. The anti-tumor effects were reversed by overexpression of a constitutively active form of Akt [12,34].

Genipin has also shown promise in oral squamous cell carcinoma (OSCC) research. It suppressed the growth of SCC-25 and SCC-9 cells and induced apoptosis and autophagy in vitro. The studies showed a decrease in p62 expression and an increase in Beclin1 and LC3II (light chain 3) expression. Genipin has also been shown to induce autophagy in OSCC cells by inhibiting the PI3K/Akt/mTOR pathway, a signaling pathway involved in cell proliferation, angiogenesis, transcription and translation [36].

Genipin also induces apoptosis in AGS and MKN45 gastric cells by suppressing the Stat3/Jak2/Mcl-1 pathway (signal transducer and activator of transcription 3/Janus kinase2/myeloid cell leukemia-1). In this case, genipin appears to contribute to the collapse of mitochondrial functions such as MMP (mitochondrial membrane potential) [37].

Genipin treatment of HCT116 colon cancer cells resulted in elevated expression of p53 and Bax, and caspase-3 cleavage, and decreased expression of Bcl-2. Genipin was able to reduce proliferation and promote apoptosis in colon cancer cells by inducing a signaling pathway mediated by p53/Bax [37].

Genipin has also been shown to reduce the metastasis and migration of hepatocellular carcinoma cells. Hong [38] report that it appears to suppress STAT-3 phosphorylation and nuclear translocation; the authors attribute this to the ability of genipin to bind to the Src homology-2 (SH2) domain of STAT-3. The STAT-3 (signal transducer and activator of transcription-3) can facilitate cancer progression and metastasis via various signaling pathways.

Tian et al. [39] reports that genipin appears to counteract the action of Fluoxetine (FXT). FXT was found to increase matrix metalloproteinase (MMPs) expression at the genetic and protein level, and to elevate urokinase-type plasminogen activator (uPA), NF-κB, activator protein 1 (AP-1), phosphorylated mitogen-activated protein kinase (P-p38) and phosphorylated protein kinase B (P-Akt) expression. It also downregulated the expression of tissue inhibitor metalloproteinase (TIMPs) at the gene and protein levels [37]. FXT is one of the top five psychiatric prescriptions in the United States and known to promote fat accumulation in primary mouse hepatocytes, suggesting it can promote the development of fatty liver, one of the causative factors of liver cancer [40].

Genipin is also thought to exert other potential anti-tumor mechanisms related to hypoxia and the expression of HIF-1 (hypoxia inducible factor 1) and VEGFR (vascular endothelial growth factor receptor), which have been found to play important roles in cancer angiogenesis progression in various cell lines. Genipin suppressed HIF-1α accumulation under hypoxia, associated with the PI3K and MAPK pathways, in human liver cancer cell line (HepG2), human prostate cancer cell line (LNCaP), colon cancer cell line (HCT116), cervical carcinoma cells, and breast cancer cell line (MDA231). Genipin also suppressed the expression of VEGF and the invasion of colon cancer cells by blocking the extracellular signal-regulated kinase signaling pathway [41,42].

Genipin may also modulate the activity of drugs used in cancer therapy. Genipin enhances the antitumor effect of elesclomol in A549 lung cancer cells by blocking uncoupling protein-2 and stimulating reactive oxygen species production [43,44]. It also has been found to inhibit the activity of mitochondrial protein UCP2. UCP2 acts as a promoter in various cancers by creating proton leaks across the inner mitochondrial membrane; by doing so, it uncouples oxidative phosphorylation from ATP synthesis, reduces the production of reactive oxygen species and the expression of matrix metalloproteinase 2, and induces caspase-dependent apoptosis in vitro and in vitro [45].

The combination of genipin and oxaliplatin has been found to exert synergistic antitumor effects in vitro and in vivo in colorectal cancer cell lines through the reactive oxygen species (ROS)/endoplasmic reticulum (ER) stress/BIM (Bcl-2-interacting mediator of cell death) pathway. The combination may offer significant therapeutic potential with minimal adverse effects [46]. Furthermore, genipin potentiates the cytotoxicity of cisplatin while simultaneously reducing markers of cisplatin-induced nephrotoxicity in a mouse- model study [12].

## 6. Other Biological Properties of Genipin

Genipin has also been found to possesses a range of other biological properties. Gupta et al. [47] report that genipin (100 µM) reduced elevated UCP2 levels in macrophages induced by infection and that it appears to possess antileishmanial potential by suppressing of UCP2 by the host. Nguyen et al. [48] suggest that a mixture of herbal preparations consisting of ginseng radix, rhei radix et rhizoma, cimicifugae rhizoma, and gareniae fructus may have therapeutic potential in the treatment of psoriasis lesions; the mixture is known to contain, among others, chlorogenic acid, emodin, genipin, cimigenoside and ginsenoside RB1. For example, at a concentration of 1%, application reduced psoriasis-like symptoms in C57BL/6 mice previously treated with imiquimod and attenuated the production of various cytokines in skin lesions. The mixture also demonstrated synergistic properties compared to its components. Moreover, genipin administration (30 mg/kg/day) also demonstrated significant suppression of liver and spleen parasite burdens in infected mice, resulting in an elevated level of p38 mitogen-activated protein kinase, a host-favorable cytokine, in a ROS-dependent manner.

Of all the active components of the *G. jasminoides* fruit extract, it has been proposed that genipin has the greatest influence on T-cell suppression, which is partially mediated by activation of the ORAI1 (ORAI calcium release-activated calcium modulator 1) channel [49]. The authors suggest that this may be the mechanism by which genipin exerts its anti-inflammatory effects. The studies were based on human primary CD4^+^T lymphocytes, with genipin concentrations ranging from 10 to 100 µM.

Genipin also demonstrated antiviral activity against human (Wa) and simian (SA-11) rotavirus strains in vitro and in vivo. The authors suggest that genipin may suppress viral replication and regulate inflammatory responses. Genipin has also shown therapeutic and preventive potential in blocking white spot syndrome virus (WSSV) by inhibiting Bax inhibitor-1 gene expression; it also inhibited WSSV replication when applied at concentrations of 6.25 to 50 mg/kg by decreasing the expression of STAT (signal transducer and activator of transcription) in crayfish and shrimp [50]. In addition, genipin glycoside derivatives have also been found to have antiviral and antifungal activities [51].

Ko et al. [52] found that genipin inhibits allergic responses in ovalbumin-induced asthmatic mice. Briefly, the mice were administrated an intraperitoneal injection of ovalbumin on days 0 and 14 to boost the immune responses, following which, genipin (10 and 20 mg/kg) was administered from day 18 to 23 by oral gavage. The authors observed that genipin significantly reduced the inflammatory cell count, the expression of inducible nitric oxide synthase and cyclooxygenase, as well as the activity and protein levels of matrix metalloproteinase-9 in the lung.

Li et al. [34] report that the PI3k/Akt pathway is involved in the mechanism of action of genipin in acute lung injury. Seo et al. [21] indicate that 25, 50, and 100 mg/kg genipin seems to prevent d-galactosamine and lipopolysaccharide-induced hepatic injury in mice through suppression of necroptosis-mediated inflammasome signaling. Elsewhere, genipin was found to also have an anti-aging effect on chondrocytes, and to demonstrate antioxidative properties in old rat hearts when administered at 5 to 10 M for 15 min before prolonged ischemia [33].

A study of spontaneously hypertensive rats found that genipin decreases blood pressure and that genipin appears to improve renal function by decreasing serum creatinine, urinary microalbumin, *N*-acetyl-β-d-glucosaminidase, and blood urea nitrogen [53]. Zhang et al. [54] suggest that genipin (1, 1.25, and 5 mg/kg) protects against LPS-induced acute lung injury by inhibiting the NFκB and NLR family pyrin domain containing 3 (NLRP3) signaling pathways in mice. In addition, 2 and 5 mg/kg genipin appears to protect against apoptosis and inflammation in LPS-induced acute lung injury by promoting autophagy [55].

Wang et al. [56] observed that genipin reduces the expression of miR-29a, miR-29b and miR-29c in the sclera of myopic eyes. The expression was determined using PCR. In addition, this compound inhibited the protein expression of matrix metalloproteinase 2 (MMP2) and increased the expression of the collagen alpha1 chain of type 1 (COL1A1); the authors suggest that genipin may be a promising agent for treating high myopia. This study was based on a model of myopia where guinea pigs were treated with a −8D lens on both eyes for 21 days. Eyes with a refractive error of −6D or greater were selected for the experiments. The scleral samples in the control group and myopia group were incubated in normal saline or in 1% genipin at 24 °C. Other experiments suggest that genipin (5–100 µM) may protect against injury induced by 100 µM or 200 µM H_2_O_2_ in retinal pigment epithelial cells via the Nrf2 signaling pathway.

Genipin (25 mg/kg) improves reproductive health problems caused by a circadian disruption in male mice. The authors observed that exogenous genipin alleviated damage to fertility and spermatogenesis induced by circadian disruption. It also normalized the levels of various hormones, including androstenedione, testosterone and dihydrotestosterone in male mice. Genipin also restored the reduction of key proteins involved in steroidogenesis.

Recently, Zhang et al. [28] report that combined genipin (50 mg/kg) and insulin (10 IU/kg) treatment improves implant osseointegration in type 2 diabetic rats, and that this may be related to the AMPK (adenosine monophosphate-activated protein kinase) signaling pathway.

The biological properties of genipin in vitro and in vivo are summarized in Table 2, and the therapeutic potential of genipin for different diseases and the main signaling molecular pathways of genipin are given in Figure 2 and Figure 3. More details about signaling pathways for genipin are described in other review papers [2,8]. However, in the absence of the identification of at least one direct cellular target of genipin (or its metabolites), its specificity of action remains vague. For example, recently, Li et al. [2] have also described that more work is required on identifying target molecules of genipin that are involved in signaling pathways that modulate neutrophic properties. Molecular targets for hepatoprotective properties of genipin are also different.

In addition, all the data reported in this review paper, on both cellular and animal models, are based on extremely high doses of genipin administered, which prompt for pharmacological application of the molecular excluding the use as a natural supplement. Therefore, a particularly important goal would be to determine appropriate therapeutic doses of genipin. Moreover, from a pharmacological point of view, future experiments should explain the broad spectrum of biological activity of genipin. They may also exclude genipin with the category of Pan-assay interference compounds (PAINS) that often give false positive results in high-throughput screens [66].

## 7. Genipin as a Crosslinking Compound

As a natural, water-soluble substance, genipin has been extensively studied as a non-cytotoxic crosslinking compound. Such crosslinking reactions between genipin and biopolymers are suitable for the food industry, pharmacological, and biomedical fields [9]. As genipin can crosslink proteins such as collagen and gelatin, and saccharides such as chitosan, it may be used for biomaterial production. Ma et al. [67] found that genipin-crosslinked chitosan can serve as a matrix for the biodegeneration of synthetic dyes. In other studies, a genipin-crosslinked human albumin coating with a tannic acid layer appeared to enhance uptake of orally administered curcumin in the treatment of ulcerative colitis. This is a very interesting experiment, because curcumin is an effective treatment but is instable in the digestive tract and has a short retention time in the colon [68]. Li et al. [34] demonstrated that genipin can act as an effective hemostatic agent by crosslinking microspheres and can enhance the stability and anti-cancer properties of curcumin by stabilizing caseinate-chitosan nanoparticles.

Genipin was found to be effective in immobilizing BMP-2 (bone morphogenetic protein-2) on roughened zirconia surfaces for enhancing cell adhesion and mineralization in dental implant applications. It also offers potential in biocatalyst design, particularly in food modifications [69]. The physicochemical characteristics of genipin-cross-linked hydrogels, as well as various information on the use of biomaterials for drug delivery, including ocular drug delivery, buccal drug delivery, anti-inflammatory drug delivery, oral drug delivery, are given in a recent review by Xu et al. [70].

## 8. Toxicity of Genipin

According to the ChemIDplus database, the LD50 genipin is 237 mg/kg (oral route) and 153–190 mg/kg (intraperitoneal route) for mice, and >50 mg/kg (oral route) for rats [7].

The toxicity of genipin has not been well investigated. However, Hoobs et al. [60] that genipin administered to mice, 74 mg/kg b.w. per day for males and 222 mg/kg b.w. for females, induces micronuclei in peripheral blood cells, nor did they cause DNA damage in the liver, duodenum, or stomach tissues. A recent study of zebrafish embryos by Xia et al. [71] found that treatment with 50 µg/mL genipin reduced hatching rates and body length, and induced nephrotoxicity, hepatotoxicity and cardiotoxicity. The authors suggest that a key role may be played by oxidative stress, particularly lipid peroxidation, measured by malondialdehyde level, and the concentration of reactive oxygen species.

However, Kim et al. [72] indicate that the blue pigment obtained from *Gardenia jasminoides* J.Ellis exhibits low cytotoxicity, and that it is effective in staining plaque, allowing it to be used as a natural dental plaque disclosant. Although the authors do not refer directly to genipin in the their study, it is known that it is the natural precursor of the pigment used in the study. Although genipin itself is colorless, it acquires its blue color by a reaction with primary amino acids and protein hydrolysates.

## 9. Conclusions

Genipin is a suitable natural blue colorant precursor for food products. Moreover, this review describes new data regarding its bioactivity, its antitumor, anti-diabetic and neuroprotective activities, and other clinically relevant topics. Particularly noteworthy is the fact that genipin demonstrates anti-cancer properties, which it exerts at all stages of the process, from proliferation, through apoptosis, to inhibition of migration and metastasis. Most importantly, in vitro and in vivo studies indicate that it is not toxic to normal cells. Hence, genipin offers promise as an effective anti-cancer agent and as a strong cross-linking drug that can be used in the development of new and effective drugs. 

Compared with results published from in vitro models, few in vivo studies have examined the biological activity of genipin and its mechanism of action remains also poorly defined. 

## Figures and Tables

**Figure 1 ijms-23-00902-f001:**
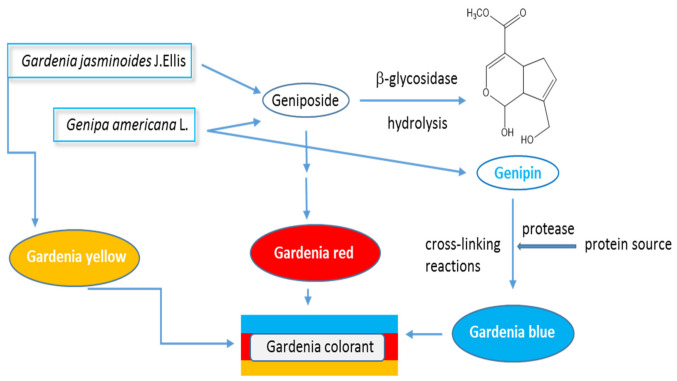
Sources of genipin and the conversion process of gardenia colorants (gardenia yellow and gardenia blue) [1].

**Figure 2 ijms-23-00902-f002:**
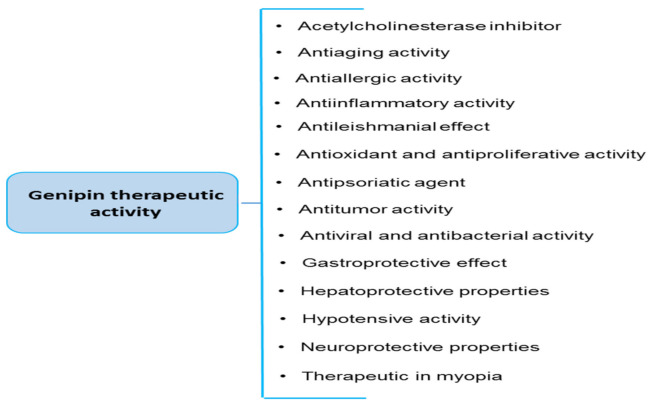
Therapeutic potential of genipin for different diseases (modified, [2]).

**Figure 3 ijms-23-00902-f003:**
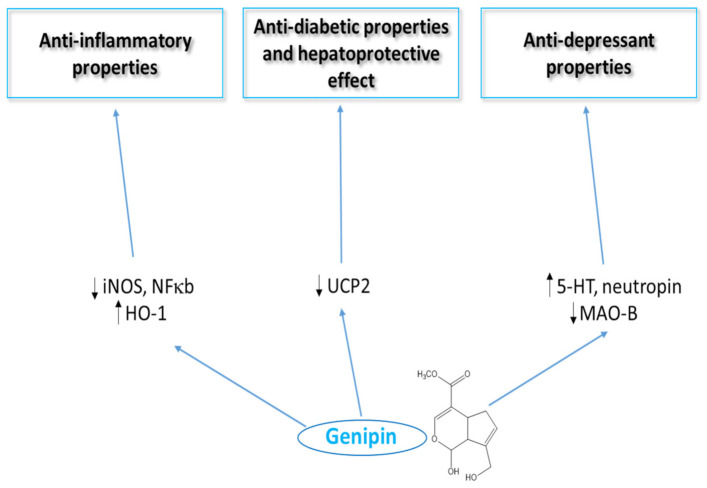
Selected signaling pathways for genipin (modified, [2,8]).

**Table 1 ijms-23-00902-t001:** Patent overview of genipin for food products [16].

Title	Publication Number and Date	Claim
Stable natural color process, products and use thereof.	USRE46695E (28 March 2008)	A method of preparing colored products from edible materials comprises processing *Genipa americana* fruit juice.
Stable natural color process and products.	WO2009120579 (6 March 2009)	Colorant for beverages, and dietary supplements.
Stable natural color comprising genipin and derivatives.	CA2718604C (15 September 2010)	A method of preparing colored products from edible materials comprises processing *Genipa americana* fruit juice.
Genipin-rich substances and uses thereof.	JP2017105851A (7 November 2011)	A method of producing a genipin rich colorant from *Genipa americana*.
A process for obtaining insoluble substances from genipap-extract precipitates, substances from genipap-extract precipitates and their uses.	EP2408319A1 (25 January 2012)	The precipitation of genipap extract (*Genipa americana*) for obtaining a substance insoluble in polar and/or non-polar media for applications in an food compositions.
Genipin-rich material and its use.	EP3238550A1 (7 November 2012) US8945640B2 (3 November 2015) USRE46314E (21 February 2017)	A method of preparing genipin-rich materials from the fruit of *Genipa americana* for their use as a cross-linking agent and as a raw material to produce colors is disclosed.
Colorant compounds derived from genipin or genipin containing materials.	US9376569 (28 June 2016) US10266698B2 (23 April 2019)	Colorant compounds and methods of its isolation from a reaction of genipin and an amine.

**Table 2 ijms-23-00902-t002:** Biological properties of genipin in various experimental models.

Compound and Its Concentration	Length of Study	Experimental Model	Biological Properties	References
Genipin (2.5–1000 µM)	-	In vitro—U87MG and A172 cell lines	Anticancer action	[57]
Geniposide and genipin (0.03–0.25 mM) Total of 31 and 62 mg/kg/day for geniposide, Total of 18 and 36 mg/kg/day for genipin	In vitro—3, 6 and 24 h In vivo—1 week	In vitro—AGS cells In vivo—C57BL/6 mice	Reducing *H. pylori* infections	[58]
Genipin (500 nM–200 µM)	24 h	In vitro—pancreatic adenocarcinoma PaCa44, PaCa3, Panc1, MiaPaCa2 and T3M4	Anticancer action	[59]
Genipin (10 and 75 µg/mL, 3–74 mg/kg bw/day)	In vitro—4 and 24 h In vivo—24 h	In vitro—human TP53 component human lymphoblast TK3 cells In vivo—B6C3F1 mice	Anticancer action	[60]
Genipin (50 mg/kg)	24 h	In vivo—red swamp crayfish *P. clarkii*	Antiviral action	[50]
Genipin (20–50 µM)	24 h	In vitro—human colorectal cancer cell lines—HCT16 and DLD-1	Therapeutic potential with a minimal adverse effect of oxaliplatin	[46]
Genipin (50 µM)	24 h	In vitro—human colon cancer lines: HCT116 and HT29, human breast cancer cell line—SKBR-3, human prostate cancer cell line—DU145	Anticancer action	[41]
Genipin (10–200 µM, 20 and 50 mg/kg/three times/week)	In vitro—48 hIn vivo—4 weeks	In vitro—human bladder cancer cells: T24 and 5637 In vivo—BALB/c (nu/nu) mice	Anticancer action	[34]
Genipap fruit extract (60.77 mg/g fdw—concentration of genipini)	-	In vitro—the tumor cell lines U251 (glioma), MCF-7 (breast), NCI-ADR/RES (breast expressing the multiple drug resistance phenotype), 786–0 (renal), NCI-H460 (lung, non-small cells), PC-3 (prostate), HT-29 (colon) and K562 (leukaemia)	Antioxidant and antiproliferative effect	[14]
Genipin (25–100 mg/kg)	-	In vivo—ICR mice	Ameliorating LPS-induced hepatocellular damage	[22]
Genipin (100 mg/kg)	-	In vivo—C57BL/6 mice	Protecting the liver from ischemia/reperfusion injury	[61]
Genipin (5–20 µM, 1–5 mg/kg)	In vitro—24 hIn vivo—3 week	In vitro—BV2 microglia cells In vivo—ICR mice	Inhibiting LPS-induced inflammatory response	[6]
Genipin (1–400 µM, 30 mg/kg)	In vitro—72 hIn vivo—week	In vitro—human tongue squamous In vivo—BALB/c nude mice	Anticancer action	[36]
Genipin (50 and 100 µM, 50 and 100 mg/kg)	-	In vitro—AGS gastric cancer cells In vivo—Sprague–Dawley rats	Gastroprotective effect	[24]
Genipin (50 mg/kg)	3 days	In vivo—C57BL/6 mice	Protecting against cerebral ischemia-reperfusion injury	[62]
Genipin (50 mg/kg)	-	In vivo—mice	Hepatoprotection against ischemia/reperfusion injury	[20]
Genipin (2.5 mg/kg)	Genipin 2 h before CCl_4_	In vivo—rats	Hepatoprotective effect in the presence CCl_4_	[21]
Genipin (5 and 20 mg/kg per day)	9 weeks	In vivo—obese mice	Alleviating hepatic lipid accumulation	[23]
Genipin (25 mg/kg)	-	In vivo—male mice	Improving reproductive health problems	[63]
Genipin (5–100 µM)	-	In vitro—retinal pigment epithelial cells	Antioxidant activity	[29]
Genipin (100 µM–in vitro; 30 mg/kg/day—in vivo)	-	In vitro—macrophages In vivo—infected mice	Antileishmanial effect	[47]
Genipin (10–200 µM—in vitro; 100 mg/kg/day—in vivo)	-	In vitro—murine macrophages RAW264.7 cells In vivo—mice	Antiviral effect	[64]
Genipin (10 and 20 mg/kg)	-	In vivo—mice	Inhibiting allergic responses	[52]
Genipin (1%)		In vitro—myopic eyes of guinea pigs	Therapeutic potential in myopia	[56]
Conjugate of genipin and tacrine	-	In vitro—SH-SY5Y cells	Inhibiting acetylcholinesterase (IC_50_ about 5.8 nM)	[27]
The mixture of herbal combinations, containing genipin (1%)	-	In vivo—C57BL/6 mice	Therapeutic potential in the treatment of psoriasis lesions	[65]
Genipin (50 mg/kg) and insulin (10 IU/kg)	-	In vivo—type 2 diabetic rats	Improving implant osseointegration	[27]

## Data Availability

Not applicable.

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
