# Peer review of "Novel Findings regarding the Bioactivity of the Natural Blue Pigment Genipin in Human Diseases"

_ijms, 2022, doi:10.3390/ijms23020902_

Round 1

Reviewer 1 Report

Review of a manuscript “Novel findings regarding the bioactivity of the natural blue pigment genipin in various diseases” by Magdalena Brys and coauthors submitted to International Journal of Molecular Sciences.

Genipin is a monoterpene present in Gardenia jasminoides which may be used as a food dye and as a medication, since it possesses anti-cancer, anti-diabetic neuroprotective and hepatoprotective activity.

The authors of the manuscript discuss its mechanism of action and possibilities of new application of genipin. This is an interesting field of study and the review will be interesting for the readers of International Journal of Molecular Sciences.

The following corrections should be made.

Title “Novel findings regarding the bioactivity of the natural blue pigment genipin in various diseases”. In the title and in the manuscript it would be beneficial if the author replace “various diseases” to “human diseases”

Introduction

Lines 26-27: After the sentence “Furthermore, preparations containing G. jasminoides fruit extract have also been found to have good therapeutic effects on diseases of the central nervous system, such as cerebral stroke, dementia, and depression (2). The authors should add the following sentence and citation: “Phytochemicals from medicinal plants often offer a better and safer substitute to synthetic medication” (Phytochemicals as Regulators of Genes Involved in Synucleinopathies. Biomolecules. 2021 11 (5):624. doi: 10.3390/biom11050624).

Lines 37-38:”a water-soluble iridoid monoterpenoid whose maximum absorbance (496 nm) does not change with the pH of the environment” The authors should explain here why this genepin property is important.

Lines 41-43:”Although various methods exist for isolating genipin from G. jasminoides fruits, large-scale isolation is usually performed microbiologically with Penicillum nigrans; the fungus hydrolyzes the geniposide into aglycone genipin by β-glucosidase (8,9).” This is confusing sentence, since microbiological method is just a first step in the large-scale isolation. The authors should briefly mentioned other following steps in the isolation and purification.

Lines 49-50:”…it is quite resistant to light radiation. It can therefore be used for coloring both liquid products and solid products, including jellies and candies…” Resistance to light radiation is not an advantage for coloring jellies and candies”.

Line 73. Genipin in food products. “Table 1 demonstrates an overview of the use of genipin in food products.” should be corrected as follows: ”The use of genipin in food products is summarized in Table 1“.

Lines 113-114: ”4. Neuroprotective properties A key element of progression in neurodegenerative diseases, such as Alzheimer’s disease and Parkinson’s disease, is the development of an inflammatory response in the brain”. The authors should add here a the citation of a new review “Biomarkers in Parkinson’s Disease. In: Peplow P.V., Martinez B., Gennarelli T.A. (eds) Neurodegenerative Diseases Biomarkers. Neuromethods, vol 173. Humana, New York, NY. https://doi.org/10.1007/978-1-0716-1712-0_7”

Line 345:”9. Conclusion”

“Genipin is not only an interesting natural blue colorant precursor for food products,…” The authors should replace here “interesting” on “convenient or suitable”

Author Response

We thank the reviewer for helpful comments. Moreover, authors agree with the comment of Reviewer, and this wrong statement was corrected.

Title “Novel findings regarding the bioactivity of the natural blue pigment genipin in various diseases”. In the title and in the manuscript it would be beneficial if the author replace “various diseases” to “human diseases”

Response: We have changed title. Now, it is “Novel findings regarding the bioactivity of the natural blue pigment genipin in human diseases”.

Introduction

Lines 26-27: After the sentence “Furthermore, preparations containing G. jasminoides fruit extract have also been found to have good therapeutic effects on diseases of the central nervous system, such as cerebral stroke, dementia, and depression (2). The authors should add the following sentence and citation: “Phytochemicals from medicinal plants often offer a better and safer substitute to synthetic medication” (Phytochemicals as Regulators of Genes Involved in Synucleinopathies. Biomolecules. 2021 11 (5):624. doi: 10.3390/biom11050624).

Response: We have added this sentence and citation.

Lines 37-38:”a water-soluble iridoid monoterpenoid whose maximum absorbance (496 nm) does not change with the pH of the environment” The authors should explain here why this genepin property is important.

Response: We have added more information about it: “Gardenia Blue is not resistant to temperatures greater than 80oC; however, it does resist changes in environmental pH. It can therefore be used for coloring both liquid products and solid products, including jellies and candies; however, dyeing jellies or candies brings with it a change of color to blue-green”.

Lines 41-43:”Although various methods exist for isolating genipin from G. jasminoides fruits, large-scale isolation is usually performed microbiologically with Penicillum nigrans; the fungus hydrolyzes the geniposide into aglycone genipin by β-glucosidase (8,9).” This is confusing sentence, since microbiological method is just a first step in the large-scale isolation. The authors should briefly mentioned other following steps in the isolation and purification.

Response: We have added the sentence: “More details are described in review paper of Ramos-de-la-Pena (2014).”

Lines 49-50:”…it is quite resistant to light radiation. It can therefore be used for coloring both liquid products and solid products, including jellies and candies…” Resistance to light radiation is not an advantage for coloring jellies and candies”.

Response: We have changed this sentence.

Line 73. Genipin in food products. “Table 1 demonstrates an overview of the use of genipin in food products.” should be corrected as follows: ”The use of genipin in food products is summarized in Table 1“.

Response: We have corrected this sentence. Now, it is “The use of genipin in food products is summarized in Table 1”.

Lines 113-114: ”4. Neuroprotective properties A key element of progression in neurodegenerative diseases, such as Alzheimer’s disease and Parkinson’s disease, is the development of an inflammatory response in the brain”. The authors should add here a the citation of a new review “Biomarkers in Parkinson’s Disease. In: Peplow P.V., Martinez B., Gennarelli T.A. (eds) Neurodegenerative Diseases Biomarkers. Neuromethods, vol 173. Humana, New York, NY. https://doi.org/10.1007/978-1-0716-1712-0_7”

Response: We have added this review paper.

Line 345:”9. Conclusion”

“Genipin is not only an interesting natural blue colorant precursor for food products,…” The authors should replace here “interesting” on “convenient or suitable”.

Response: We have changed this sentence. Now, it is: “Genipin is a suitable natural blue colorant precursor for food products.”

Reviewer 2 Report

Although the present review is well written and easy to follow, I do not think that it is adequate for the qualitative standards expected by the IJMS readers. The main weaknesses that I identified are listed below:

-Limited novelty. The same authors published very recently (Adv Nutr. 2021 Dec 1;12(6):2301-2311. doi: 10.1093/advances/nmab081) a review article that covers arguments and concepts repeated in the present submission. I don’t see any significant “new findings”, as reported in the title, regarding the bioactivity of genipin.

-The present work suffers of the weakness common to similar articles dealing with the broad spectrum of biological activities of natural compounds, e.g. the specificity of the molecular mechanism of action. It is not credible from a pharmacological point of view that the same compound can show therapeutic efficacy against so many and so different diseases from a patho-physiological point of view as those listed in Fig. 2. On the opposite, I tend to believe that genipin can fall within the category of PAINS (pan-assay interference compounds. See specific literature on this issue). The authors should convincingly exclude this possibility.

-Essentially all the data reported in the present work, on both cellular and animal models, are based on extremely high doses of genipin administered, which prompt for pharmacological application of the molecule excluding the use as supplements. Not enough information is reported to discriminate between these two different fields of application.

-I did not perceive any attempt of the authors to propose one or more mechanism(s) of action that, at the molecular level, can explain the multiple effects of genipin. This represents, in my view, a profound weakness considering that reading the paragraphs, the article suggests the existence of potential, preferred molecular targets of genipin.

-The manuscript ends with a Conclusion paragraph without a section for “Discussion” where the authors could have constructively and critically expressed their opinions and suggestions on future directions in the field.

Author Response

We thank the reviewer for helpful comments. Moreover, authors agree with the comment of Reviewer, and this wrong statement was corrected.

-Limited novelty. The same authors published very recently (Adv Nutr. 2021 Dec 1;12(6):2301-2311. doi: 10.1093/advances/nmab081) a review article that covers arguments and concepts repeated in the present submission. I don’t see any significant “new findings”, as reported in the title, regarding the bioactivity of genipin.

Response: Our earlier review paper (Olas et al., 2021; Adv Nutr) describes the effects of various natural and synthetic blue dyes on human health, including only short chapter about biological properties of genipin. This present review paper describes not only bioactivity of genipin, but also its toxicity, industrial applications, and its potential mechanism of action. This review is based on studies identified in electronic databases, including PubMed, Web of Knowledge, Scopus and ScienceDirect. The last search was run on December 15, 2021. The following terms were used: “genipin” and “blue colorant”. We have added this information (chapter of Introduction).

-The present work suffers of the weakness common to similar articles dealing with the broad spectrum of biological activities of natural compounds, e.g. the specificity of the molecular mechanism of action. It is not credible from a pharmacological point of view that the same compound can show therapeutic efficacy against so many and so different diseases from a patho-physiological point of view as those listed in Fig. 2. On the opposite, I tend to believe that genipin can fall within the category of PAINS (pan-assay interference compounds. See specific literature on this issue). The authors should convincingly exclude this possibility.

Response: We have added more information about it: “The biological properties of genipin in vitro and in vivo are summarized in Table 2, and the therapeutic potential of genipin for different diseases and the main signaling molecular pathways of genipin are given in Figure 2. On the other hand, genipin can fall within the category of Pan-assay interference compounds (PAINS). PAINS are chemical compounds that often give false positive results in high-throughput screens (Baell and Nissnik, 2018).”

-Essentially all the data reported in the present work, on both cellular and animal models, are based on extremely high doses of genipin administered, which prompt for pharmacological application of the molecule excluding the use as supplements. Not enough information is reported to discriminate between these two different fields of application.

Response: We have added more information about it (chapter of Conclusion): “Genipin is a suitable natural blue colorant precursor for food products. Moreover, this review describes new data regarding its bioactivity, its antitumor, anti-diabetic and neuroprotective activities, and other clinically-relevant topics. Particularly noteworthy is the fact that genipin demonstrates anti-cancer properties, which it exerts at all stages of the process, from proliferation, through apoptosis, to inhibition of migration and metastasis. Most importantly, in vitro and in vivo studies indicate that it is not toxic to normal cells. Hence, genipin offers promise as an effective anti-cancer agent and as a strong cross-linking drug that can be used in the development of new and effective drugs. On the other hand, all the data reported in this review paper, on both cellular and animal models, are based on extremely high doses of genipin administered, which prompt for pharmacological application of the molecular excluding the use as a natural supplement. Therefore, a particularly important goal would be to determine appropriate therapeutic doses of genipin.”.

-I did not perceive any attempt of the authors to propose one or more mechanism(s) of action that, at the molecular level, can explain the multiple effects of genipin. This represents, in my view, a profound weakness considering that reading the paragraphs, the article suggests the existence of potential, preferred molecular targets of genipin.

Response: We have added more information about signaling pathways for genipin: “The biological properties of genipin in vitro and in vivo are summarized in Table 2, and the therapeutic potential of genipin for different diseases and the main signaling molecular pathways of genipin are given in Figure 2 and 3. On the other hand, genipin can fall within the category of Pan-assay interference compounds (PAINS). PAINS are chemical compunds that often give false positive results in high-throughput screens (Baell and Nissnik, 2018). More details about signaling pathways for genipin are described in other review papers (2 and 7).”. Moreover, we have added new figure – Fig. 3.

-The manuscript ends with a Conclusion paragraph without a section for “Discussion” where the authors could have constructively and critically expressed their opinions and suggestions on future directions in the field.

Response: We have changed Conclusion.

Round 2

Reviewer 2 Report

I thank the authors for their replies. However, the limited changes provided to the manuscript, in my opinion, do not significantly improve its quality. On the opposite, in some cases, they worsened the significance of their work. As an example, confirming that “genipin can fall within the category of Pan-assay interference compounds (PAINS)”, the authors are implicitly admitting that the biological activities of genipin can be attributed to artifacts. About the mechanism(s) of action, in the absence of the identification of at least one direct cellular target of genipin (or its metabolites), its specificity of action remains vague. Finally, I do not think that the level of novelty of the present goes significantly far away from the data reported in Olas et al., 2021.

Author Response

I thank the authors for their replies. However, the limited changes provided to the manuscript, in my opinion, do not significantly improve its quality. On the opposite, in some cases, they worsened the significance of their work. As an example, confirming that “genipin can fall within the category of Pan-assay interference compounds (PAINS)”, the authors are implicitly admitting that the biological activities of genipin can be attributed to artifacts. About the mechanism(s) of action, in the absence of the identification of at least one direct cellular target of genipin (or its metabolites), its specificity of action remains vague. Finally, I do not think that the level of novelty of the present goes significantly far away from the data reported in Olas et al., 2021

Response:

We thank the reviewer for helpful comments. Authors agree with the comment of Reviewer, and this wrong statement was corrected.

We have added more information: ”The biological properties of genipin in vitro and in vivo are summarized in Table 2, and the therapeutic potential of genipin for different diseases and the main signaling molecular pathways of genipin are given in Figure 2 and 3. More details about signaling pathways for genipin are described in other review papers (2 and 8). However, in the absence of the identification of at least one direct cellular target of genipin (or its metabolites), its specificity of action remains vague. For example, recently, Li et al. (2) have also described that more work is required on identifying target molecules of genipin that are involved in signaling pathways that modulate neutrophic properties. Molecular targets for hepatoprotective properties of genipin are also different.

In addition, all the data reported in this review paper, on both cellular and animal models, are based on extremely high doses of genipin administered, which prompt for pharmacological application of the molecular excluding the use as a natural supplement. Therefore, a particularly important goal would be to determine appropriate therapeutic doses of genipin. Moreover, from a pharmacological point of view, future experiments should explain the broad spectrum of biological activity of genipin. They may also exclude genipin with the category of Pan-assay interference compounds (PAINS) that often give false positive results in high-throughput screens (59).”

Our earlier review paper (Olas et al., 2021; Adv Nutr) describes the effects of various natural and synthetic blue dyes on human health, including only short chapter about selected biological properties of genipin. This present review paper describes not only bioactivity of genipin, but also its toxicity, industrial applications, and its potential mechanism of action. This review is based on studies identified in electronic databases, including PubMed, Web of Knowledge, Scopus and ScienceDirect. The last search was run on December 15, 2021. The following terms were used: “genipin” and “blue colorant”.